# Sigmoidally hydrochromic molecular porous crystal with rotatable dendrons

Hiroshi Yamagishi [1], Sae Nakajima[1], Jooyoung Yoo[1], Masato Okazaki[2], Youhei Takeda [2✉], Satoshi Minakata[2], Ken Albrecht [3,4,5✉], Kimihisa Yamamoto[3,4], Irene Badía-Domínguez [6], Maria Moreno Oliva[6], M. Carmen Ruiz Delgado[6], Yuka Ikemoto[7], Hiroyasu Sato[8], Kenta Imoto[9], Kosuke Nakagawa [9], Hiroko Tokoro [1], Shin-ichi Ohkoshi [9] & Yohei Yamamoto [1✉]

Vapochromic behaviour of porous crystals is beneficial for facile and rapid detection of gaseous molecules without electricity. Toward this end, tailored molecular designs have been established for metal–organic, covalent-bonded and hydrogen-bonded frameworks. Here, we explore the hydrochromic chemistry of a van der Waals (VDW) porous crystal. The VDW porous crystal **VPC-1** is formed from a novel aromatic dendrimer having a dibenzophenazine core and multibranched carbazole dendrons. Although the constituent molecules are connected via VDW forces, **VPC-1** maintains its structural integrity even after desolvation. **VPC-1** exhibits reversible colour changes upon uptake/release of water molecules due to the charge transfer character of the constituent dendrimer. Detailed structural analyses reveal that the outermost carbazole units alone are mobile in the crystal and twist simultaneously in response to water vapour. Thermodynamic analysis suggests that the sigmoidal water sorption is induced by the affinity alternation of the pore surface from hydrophobic to hydrophilic.

[1] Department of Materials Science, Faculty of Pure and Applied Sciences, and Tsukuba Research Center for Energy Materials Science (TREMS), University of Tsukuba, 1-1-1 Tennodai, Tsukuba, Ibaraki 305-8573, Japan. [2] Department of Applied Chemistry, Graduate School of Engineering, Osaka University, 2-1 Yamadaoka, Suita, Osaka 565-0871, Japan. [3] Laboratory for Chemistry and Life Science, Tokyo Institute of Technology, 4259 Nagatsuta Midori-ku, Yokohama 226-8503, Japan. [4] ERATO Yamamoto Atom Hybrid Project, Japan Science and Technology Agency (JST), 4259 Nagatsuta Midori-ku, Yokohama 226-8503, Japan. [5] Institute for Materials Chemistry and Engineering, Kyushu University, 6-1 Kasuga-koen, Fukuoka 816-8580, Japan. [6] Department of Physical Chemistry, University of Malaga, Campus de Teatinos s/n, 29071 Malaga, Spain. [7] Japan Synchrotron Radiation Research Institute (JASRI) SPring-8, 1-1-1 Koto, Sayo, Hyogo 679-5198, Japan. [8] Rigaku Corporation, 12-9-3 Matsubara, Akishima, Tokyo 196-8666, Japan. [9] Department of Chemistry, School of Science, The University of Tokyo, 7-3-1 Hongo, Bunkyo-ku, Tokyo 113-0033, Japan. ✉email: takeda@chem.eng.osaka-u.ac.jp; albrecht@cm.kyushu-u.ac.jp; yamamoto@ims.tsukuba.ac.jp

Vapochromism is an ability of chromophoric compounds to alter their colour depending on the chemical composition of the surrounding atmosphere. Typically, upon adsorption/desorption of vapour molecules, vapochromic compounds change its colours as a response towards the guest-induced change in polarities of the medium or the guest-induced rearrangement of the molecular conformation[1–7]. Especially in the former case, colour change occurs as a result of the changes in the electronic state: The ground and excited states of the vapochromic molecules experience energetic stabilization by the dipole–dipole interaction with the gaseous molecules, leading to widening/narrowing of the energy gap[8]. In principle, peak positions of the absorption and fluorescence spectra are in a linear relationship with the permittivity of the medium molecules, as theoretically derived by Lippert and Mataga[9]. An empirical parameter for the guest polarity ($E_T(30)$) has also been invented and successfully applied to diverse chromic molecules[8].

An established way to enhance the sensitivity of the chromic properties in the solid state is to form a microporous structure with a large surface area. An authentic example was reported by Zheng and co-workers in 2011 on a solvatochromic behaviour of bimetallic nanotubular metal–organic framework (MOF)[7]. The porous architecture allows adsorption of guest molecules with high efficiency, which is advantageous for rapid detection of the guest molecules. Along this line, the scope of the materials has been expanded to covalent–organic frameworks (COFs) and hydrogen-bonded organic frameworks (HOFs)[10–14].

A challenging yet practically valuable target in this field is water. The detection of water contents in the atmosphere by vapochromism (hydrochromism) is essential, especially where the electrical detection cannot be adopted. A traditional choice for this purpose is the cobalt(II) complexes that display visible colour change due to the coordination of the water molecules to the cobalt(II) centre. On the other hand, MOFs, COFs, and HOFs are generally inappropriate to this end because of their instability against water[15–18]. Chemical bonds that underpin their porous frameworks, that is, coordination bonds, dynamic covalent bonds, and hydrogen bonds, are commonly affected by the water molecules. Elaborate molecular design is required to drastically enhance the water resistance of those chemical bonds. One would assume that non-polar bonds, such as van der Waals (VDW) interaction might be an alternative strategy. However, the synthesis and engineering of VDW porous crystals is still in a primitive stage[19,20]. Actually, only several examples of VDW porous crystals have been reported so far[21–26]. Functionalization of the VDW porous crystals remains a far more challenging topic because even a slight modification of the constituent molecule or crystallization conditions ends up with the formation of densely packed polymorphs[24].

Herein, we report a hydrochromic VDW porous crystal. The hydrochromic VDW porous crystal VPC-1 consists of a newly synthesised second-generation dendrimer 1 having a planar dibenzo[a,j]phenazine (DBPHZ) core with two branched carbazole (Cz) dendrons. VPC-1 features a robust crystalline framework with permanent microporosity that exhibits sigmoidal $H_2O$ uptake/release without hysteresis. The $H_2O$ uptake/release triggers visible colour change between red and yellow with a steep threshold. Spectroscopic and X-ray diffractometry studies reveal the synchronous twisting of the outermost Cz units of 1 upon $H_2O$ adsorption/desorption, while its porous crystalline lattice neither expands, contracts nor deforms in this process. A thermodynamic analysis of the isotherms discloses the affinity transition of the pore surface from hydrophobic to hydrophilic, which is responsible for the simultaneous twisting of the Cz units.

## Results and discussion
**Synthesis of 1 and VDW porous crystal VPC-1.** The microporous crystal VPC-1 consists of a newly synthesised second-generation fully aromatic dendrimer 1 having an electron-deficient DBPHZ core and two electron-rich Cz dendrons (Fig. 1a). The DBPHZ core and Cz dendrons were separately synthesised according to the previous literatures[27–29] and, thereafter, were connected together through a Pd-catalysed Hartwig-Buchwald amination reaction. The molecular structure of 1 was unambiguously characterized by NMR spectroscopy, mass spectrometry, and single-crystal X-ray structure analysis (Fig. 2a, b, Supplementary Figs. 1–5 and Supplementary Table 1). Single crystals of 1 were grown as follows: A MeOH layer containing dispersed solid powder of 1 was gently put on a $CHCl_3$ solution of 1 (1 mg mL$^{-1}$). The container was incubated for three days at 25 °C to yield yellow-coloured platelet single crystals with side length of several tens of micrometres (note: the crystal structure is not identical with VPC-1). In the crystal structure, four crystallographically non-equivalent pairs of 1 are packed together in a unit cell with a space group of $P\bar{1}$. These molecules display twisting of the internal Cz units relative to the DBPHZ plane with dihedral angles ($\theta_1$) ranging from 46 to 55°, while the dihedral angles between the internal and external Cz ($\theta_2$) ranges from 52 to 90° (Fig. 2a). The crystal structure indicates that the external Cz units have higher degree of rotational freedom than the internal Cz units. Supplementary Fig. 6a shows the electronic absorption spectrum of 1 in $CHCl_3$ (orange curve). Therein, two sets of absorption bands emerge with peak tops at 431 and 327 nm, which derive from a charge transfer (CT) transition from Cz to DBPHZ and a π–π* transition of Cz, respectively[29]. A $CHCl_3$ solution of 1 shows photoluminescence with a unimodal emission band centred at 582 nm (Ex. 430 nm, Supplementary Fig. 6a). The CT nature was verified by $E_T30$[8] and Lippert-Mataga

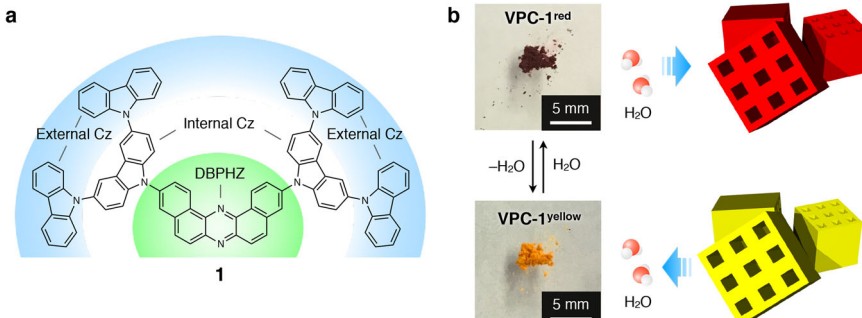

**Fig. 1 Schematic representations of the hydrochromism of VPC-1 upon hydration/dehydration. a** Molecular structure of a second-generation dendrimer **1** with two branched Cz dendrons and a DBPHZ core. **b** Photographs of **VPC-1**$^{red}$ and **VPC-1**$^{yellow}$ together with schematic illustrations of the microporous crystalline grains that change their colour in response to $H_2O$ vapour.

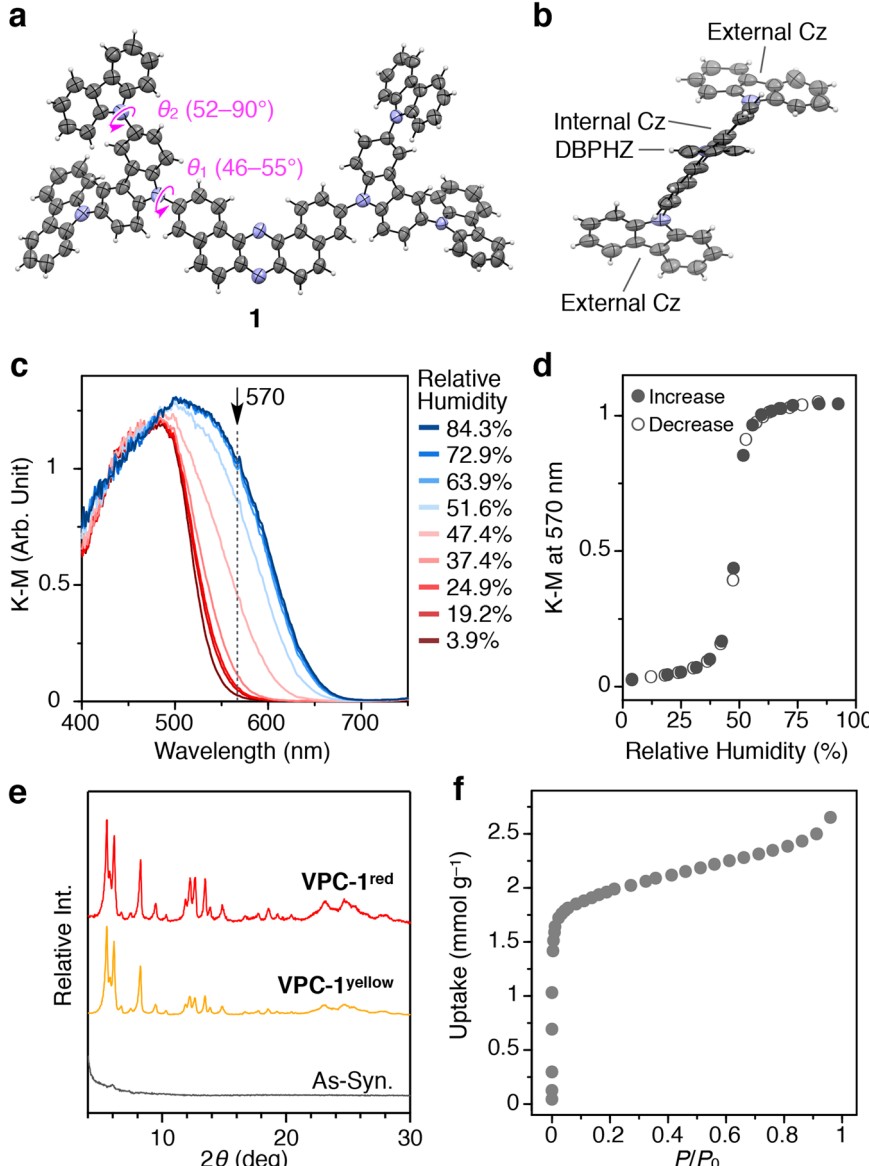

**Fig. 2 ORTEP diagrams of 1 and characterizations of VPC-1. a** An ORTEP diagram of one of the four crystallographically non-equivalent molecules of **1** with a probability level of 30%. Pink arrows indicate the twisting of the internal Cz unit with respect to the DBPHZ core and the external Cz unit with dihedral angles $\theta_1$ and $\theta_2$, respectively. **b** A side-view of one of the four crystallographically non-equivalent molecules of **1** with a probability level of 30%. **c** Humidity-dependent diffuse reflectance spectra of **VPC-1** upon increasing the surrounding humidity from 3.9 to 84.3%. **d** Plots of the K–M values at 570 nm upon increasing and decreasing the relative humidity. **e** Powder X-ray diffraction profiles of **VPC-1red** (red), **VPC-1yellow** (orange), and as-synthesised amorphous yellow solid (grey). **f** $N_2$ adsorption isotherm of **VPC-1** at −196 °C.

(LM) plots[9]. The absorption band at 431 nm shifts slightly towards the shorter wavelength when dissolved in more polar solvents with a positive linear correlation in the $E_T$30 plot (Supplementary Fig. 6b). Likewise, the emission band (Ex. 430 nm) shifts monotonically towards longer wavelength in proportion to the solvent polarity with a linear correlation in the LM plot (Supplementary Fig. 6c).

The microfibrous crystalline powder of **1** (**VPC-1**) was obtained by the solvent-assisted thermal annealing method. As-synthesised amorphous yellow solid of **1** was suspended in a mixture of $CHCl_3$ and MeOH (3/7 v/v) at 60 °C for 2 h. After the subsequent incubation at 25 °C for 3 days, the colour of the precipitates eventually turned from yellow to red. This colour change is associated with a phase change from amorphous to crystalline state as revealed by the powder X-ray diffraction (PXRD) patterns (Fig. 2e, grey and red curves, respectively).

Unfortunately, the resultant crystalline fibres were inapplicable to the single-crystal X-ray structure analysis due to its high polycrystallinity.

In the course of these experiments, a vapochromic behaviour of **VPC-1** was found (Fig. 1b). The red crystalline powder **VPC-1** readily turned into yellow upon drying under a reduced pressure (Supplementary Movie 1), whereas it reverted back into the initial red-coloured solid upon exposure to humid air (hereafter, the red and yellow powder of **VPC-1** are denoted as **VPC-1red** and **VPC-1yellow**, respectively, and specimens that include both **VPC-1red** and **VPC-1yellow** are denoted as **VPC-1**). The complete desolvation of **VPC-1** was further confirmed by thermogravimetric analysis and differential scanning calorimetry measurement under constant Ar flow condition (Supplementary Figs. 9 and 10). Consistently, diffuse reflectance spectra of **VPC-1** exhibit a bathochromic shift of the CT band from 482 to 501 nm upon

increasing the surrounding relative humidity (RH) from 3.9 to 84.3% (Fig. 2c). The plot of the K–M values at 570 nm displays a sigmoidal curve with a threshold at a RH around 50% (Fig. 2d). The dehydration process exhibits backward peak shift at a virtually identical threshold (Fig. 2d, Supplementary Fig. 8). The $E_T30$ plot of the CT bands of **VPC-1** in the diffuse reflectance spectra displays a linear correlation against the solvent polarity (Supplementary Fig. 7), which indicates that the colour of **VPC-1** is affected by the high permittivity of the guest $H_2O$ molecules as is usual with vapochromic porous crystals[6,7,11]. Of particular note, no noticeable peak shift was observed in the PXRD pattern through the colour change, indicating that the crystals preserve its lattice structure upon uptake/release of $H_2O$ molecules (Fig. 2e, red and orange curves). The colour change between **VPC-1**$^{yellow}$ and **VPC-1**$^{red}$ was completed within ten seconds upon drying under reduced pressure, which is much less than the interval time required for the stabilization of humidity or temperature (Supplementary Movie 1).

We conducted variable-humidity (VH) and variable-temperature (VT) PXRD measurement with **VPC-1** to evaluate the detailed structural change upon $H_2O$ sorption. The VH-PXRD patterns at 22.6 °C exhibited an abrupt and reversible enhancement/attenuation of the peak area upon increasing and decreasing the humidity with a threshold at around 40% RH, while no significant peak shift was observed (Supplementary Fig. 11). The unchanged peak positions indicate that the crystalline lattice of **VPC-1** is static upon $H_2O$ sorption, while the enhancement/attenuation of the peak area indicates the decreased/increased degree of the thermal disorder of **1**. Analogously, VT-PXRD patterns, measured in an unsealed chamber under atmosphere with RH of 69% at 25 °C, exhibited abrupt and reversible peak enhancement and attenuation with a threshold at around 33 °C without any noticeable peak shift upon heating and cooling (Supplementary Fig. 12). The threshold temperatures decreased as the humidity of the surrounding atmosphere decreased (Supplementary Figs. 13 and 14). The elevated temperature triggers the decrease of the RH in the sample chamber and the release of $H_2O$ molecules from **VPC-1**$^{red}$, leading to the enhancement of the peak intensity in the PXRD profiles. Based on the VH- and VT-PXRD results, a relative humidity–temperature phase diagram for **VPC-1**$^{yellow}$ and **VPC-1**$^{red}$ is depicted (Supplementary Fig. 15).

We found that **VPC-1**$^{yellow}$ is a microporous crystal that can accommodate guest gaseous molecules inside it. Nitrogen adsorption isotherm of **VPC-1**$^{yellow}$ at −196 °C exhibits an abrupt uptake in the low coverage region with a total uptake volume of 2.65 mmol $g^{-1}$ and a Brunauer–Emmett–Teller (BET) surface area of 112 $m^2 g^{-1}$ (Fig. 2f). A pore diameter distribution analysis based on the Horvath–Kawazoe (HK) model[27] displayed a skewed profile with a maximum peak at 5.6 Å (Supplementary Fig. 16). The permanent microporosity of **VPC-1**$^{yellow}$ itself is worth highlighting because the engineering of porous VDW crystals is a fundamental challenge[19,20]. This fact prompted us to investigate the detailed molecular configuration of **1** in **VPC-1** and its $H_2O$ absorption behaviour by spectroscopic and computational methods.

**Simultaneous twisting of the external Cz**. For the investigation of the molecular conformations of **1** in **VPC-1**$^{yellow}$ and **VPC-1**$^{red}$, FTIR and Raman spectroscopies were conducted, along with the DFT calculations. The FTIR spectra were measured with a pre-dried powder sample of **VPC-1**$^{yellow}$ in a humidity-control cell at 25 °C (Fig. 3a–d, for the details, see 'Methods' section). While most of the absorption bands in the fingerprint region were insensitive to the change in RH, those at 1138, 1273, and 1343 $cm^{-1}$ exhibit an abrupt band shift or attenuation when RH

increased to 56.7% (Fig. 3b, c). Concomitantly, the colour of the specimen turned from yellow to red. These spectral changes unambiguously suggest the conformational change of the constituent **1** upon $H_2O$ adsorption. Coincidentally, a broad absorption band centred at 3448 $cm^{-1}$ (O–H stretching mode of $H_2O$) emerged, which is a direct evidence of the uptake of $H_2O$ molecules (Fig. 3d). The transmittances at 1142 and 3448 $cm^{-1}$ were respectively plotted against RH, giving stepped profiles with an identical threshold between 49.7 and 56.7% (Fig. 3g, open and closed circles). Reverse spectral changes were observed in the dehydration process, showing a threshold between 59.4 and 52.0% (Supplementary Fig. 17). The observed threshold is virtually identical to that observed in the humidity-dependent diffuse reflectance spectroscopy (Fig. 2c). Meanwhile, Raman spectra of **VPC-1** measured with a 785 nm-laser in the same humidity control cell did not show any noticeable spectral change at upon increasing the relative humidity (Supplementary Fig. 18).

Computational simulations by using CAM-B3LYP[30] functional and the 6-31G(d,p)[31] basis set as implemented in the Gaussian 16 package[32] provide structural models of **1** that account for the observed IR and Raman spectral features (Supplementary Fig. 19, for details, see 'Methods' section). According to the calculations, the band at 1138 $cm^{-1}$ is attributed to the C–H bending modes on the central DBPHZ core and the adjacent internal Cz units, while the newly appeared band at 1142 $cm^{-1}$ is attributed to the C–H bending modes at the internal and external Cz units (Supplementary Fig. 20). The band at 1273 $cm^{-1}$ is likewise assigned to the C–H bending of the internal Cz and the breathing of the pyrrole moiety in the internal Cz units. On the other hand, the peak positions and intensities of Raman bands are sensitive to the twisting of the internal Cz units relative to the central DBPHZ unit (Supplementary Fig. 21). The simulated molecular conformations of **1** in **VPC-1**$^{yellow}$ and **VPC-1**$^{red}$ are shown in Fig. 3e, f. In **VPC-1**$^{yellow}$, the dihedral angle between the external and internal Cz units ($\theta_2$) are almost perpendicular (~90°), while that in **VPC-1**$^{red}$ becomes less twisted with $\theta_2$ of ~60°. In contrast, the dihedral angles between the internal Cz units and the DBPHZ core ($\theta_1$) are kept constant for both **VPC-1**$^{yellow}$ and **VPC-1**$^{red}$ (~30°). Altogether, the vibrational spectral features of **VPC-1**$^{yellow}$ and **VPC-1**$^{red}$ show that the twisting of the outermost Cz units are responsive to the surrounding humidity while the remaining moieties of **1** were intact. Taking this fact and the results of PXRD measurements, we conclude that the external Cz units are solely responsive and mobile in **VPC-1**$^{yellow}$ and **VPC-1**$^{red}$, while the other moiety of **1** and the crystal packing mode are static in the $H_2O$ sorption process. This is in clear contrast with the conventional flexible porous crystals, which transform the whole crystalline lattices and the conformation of the constituent molecules simultaneously upon guest sorption.

The electronic absorption spectra for the conformers of **1** in **VPC-1**$^{yellow}$ and **VPC-1**$^{red}$ were likewise simulated based on the standard TD-DFT calculations. As expected, the coplanarisation of the Cz shells enhances CT from Cz units to the DBPHZ core, thereby inducing the red-shift of its electronic absorption band (Supplementary Figs. 22–24). Thus, we conclude that the above-described colour change of **VPC-1** from yellow to red is enhanced by the coplanarisation of the Cz shells along with the bathochromic solvatochromism.

**Sigmoidal $H_2O$ sorption behaviours of VPC-1**. The abrupt uptake/release of $H_2O$ molecules was assessed in detail by $H_2O$ isotherms (Fig. 4a–c). A pre-dried powder sample of **VPC-1**$^{yellow}$ was put in a glass tube and subjected to the vapour isotherm measurements. The uptake of $H_2O$ ($n_{H2O}$) was negligible in the low coverage region ($n_{H2O} < 0.63$ mmol $g^{-1}$, $P/P_0 < 0.34$). On the

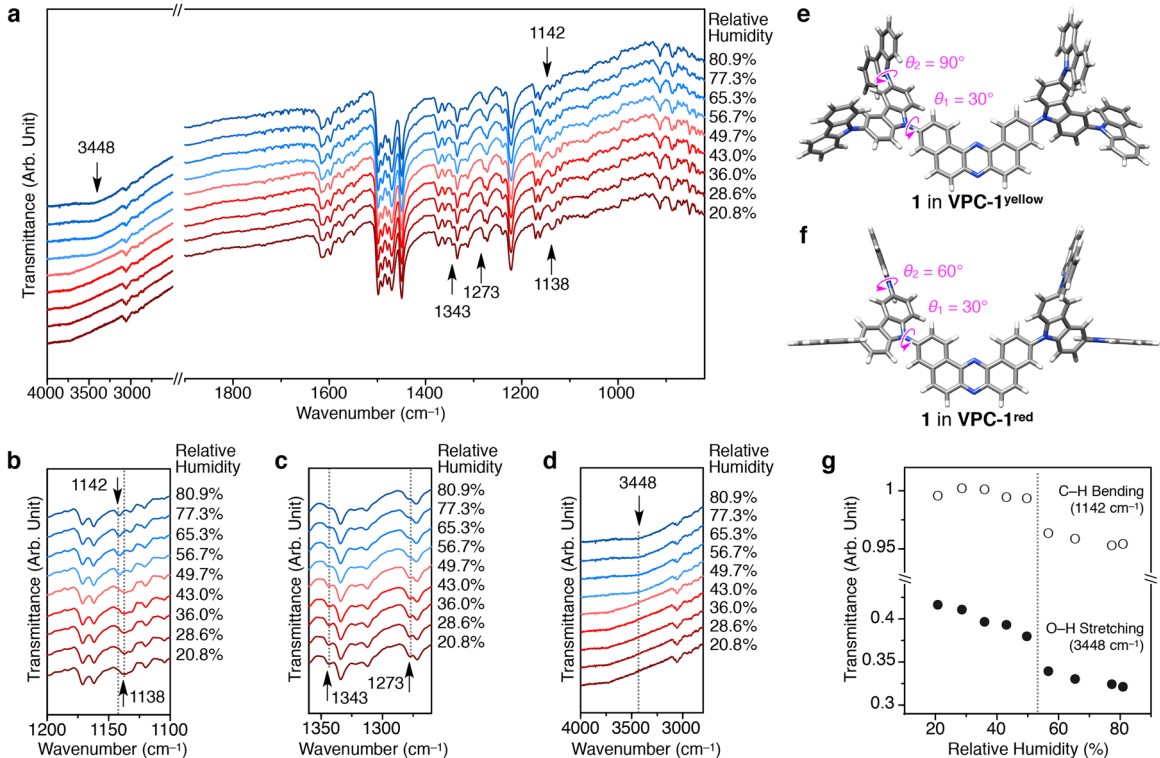

**Fig. 3 Spectroscopic data and computational molecular models for the simultaneous twisting of the external Cz units in response to H₂O uptake.**
**a–d** FTIR spectra of **VPC-1** on increasing the relative humidity at 25 °C. Selected spectral regions including vibrational bands attributed to C–H bending (**b**), C–N stretching, pyrrole breathing and C–H bending (**c**) and O–H stretching modes (**d**) are magnified for clarity. **e, f** Proposed molecular structures of **1** in **VPC-1yellow** (**e**) and **VPC-1red** (**f**), which are computationally modelled based on the FTIR, Raman and diffuse reflectance spectra. **g** Changes in the transmittance of the C–H bending (open circles) at 1142 cm⁻¹ and O–H stretching (closed circles) vibrational bands at 3448 cm⁻¹ upon increasing RH from 20.8 to 80.9%.

other hand, H₂O vapour with $n_{H_2O}$ of 1.35 mmol g⁻¹ was abruptly adsorbed in a relative pressure range from 0.34 to 0.41 ($n_{H_2O}$ = 0.63–1.98 mmol g⁻¹) with an inflection point at 0.37 ($n_{H_2O}$ = 1.35 mmol g⁻¹, 1.7 H₂O molecules per one molecule of **1**). As further increase of the relative pressure, **VPC-1** adsorbed H₂O molecules sluggishly up to 3.64 mmol g⁻¹, that is, 4.6 H₂O molecules per one molecule of **1**. The total H₂O uptake volume (0.0657 mL g⁻¹) is in reasonable agreement with the pore volume of **VPC-1yellow** (0.0865 mL g⁻¹) obtained from the N₂ adsorption isotherm, implying that the porous architecture neither expand nor shrink by the simultaneous twisting of the external Cz units of **1**. The desorption profile virtually traces the adsorption curve with a threshold relative pressure of 0.36 (Fig. 4b, open circles). Such a non-hysteretic sorption behaviour indicates the reversible and continuous pore filling by H₂O molecules rather than a capillary condensation[15,33]. Notably, vapochromic materials without hysteresis are highly desirable for chemical sensing but have never been achieved with VDW porous crystals. The threshold pressure observed in the isotherm ($P/P_0$ = 0.35–0.38) is smaller than the threshold relative humidity observed in the FTIR spectra (RH = 52–57%). This difference is presumably because all the above-described spectroscopic measurements were conducted under N₂ atmosphere that contain only a minute portion of H₂O vapour up to 3.1% at 25 °C, whereas the sorption isotherm is measured in a sealed chamber filled with H₂O vapour. As reported previously, the combination of non-hysteresis, steepness and position of the sigmoidal curve in Fig. 4a–c is appealing for the water harvesting[34]. Such ideal sorption properties have not yet been achieved with the VDW porous crystals, while zirconium- and aluminum-based porous metal–organic frameworks have been demonstrated to work well[35].

A mechanistic insight into the sigmoidal sorption was provided by the isosteric heat of adsorption ($Q_{st}$), calculated based on the H₂O isotherms of **VPC-1** measured at 10, 20 and 30 °C (Fig. 4a–c). The adsorption and desorption profiles at each temperature are essentially identical with one another with a slight monotonic shift of the threshold pressure upon increasing temperature. As shown in Fig. 4d, the plot of $Q_{st}$ against the molar amount of H₂O uptake ($n_{H_2O}$) was calculated from those isotherms by using the Clausius–Clapeyron Eq. (1),

$$\ln \frac{p_{T_2}}{p_{T_1}} = \frac{\Delta H}{R}\left(\frac{1}{T_1} - \frac{1}{T_2}\right) \qquad (1)$$

At a lower H₂O coverage, the $Q_{st}$ plot displays a positive slope until $n_{H_2O}$ reaches 0.61 mmol g⁻¹ (0.78 mol per 1 mol of **1**), which corresponds to the beginning of the step region in the H₂O isotherm measured at 20 °C. In the following region ($n_{H_2O}$ = 0.61–1.93 mmol g⁻¹), the $Q_{st}$ plot exhibits a plateau with $Q_{st}$ of 44 ± 1 kJ mol⁻¹, which is a typical value for H₂O physisorption to the hydrophilic surface. Thereafter, $Q_{st}$ gradually decreases to 37 kJ mol⁻¹. The noticeable enhancement of $Q_{st}$ in the lower coverage region below the threshold indicates that the initial hydrophobic pore surface of **HCF-1yellow** gradually becomes hydrophilic as the humidity increase. We attribute this affinity transition to the twist-induced structural change of the pore surface. The calculated charge distribution for an isolated carbazole unit shows that the nitrogen atom in the carbazole unit bears the largest amount of negative charge, while the other part of the Cz unit is less charged (Supplementary Fig. 25). Therefore, it is most likely that, in the small H₂O pressure region, the fused benzene part, which is originally hydrophobic, is exposed in the pore surface. Due to this hydrophobicity, only a small portion of H₂O

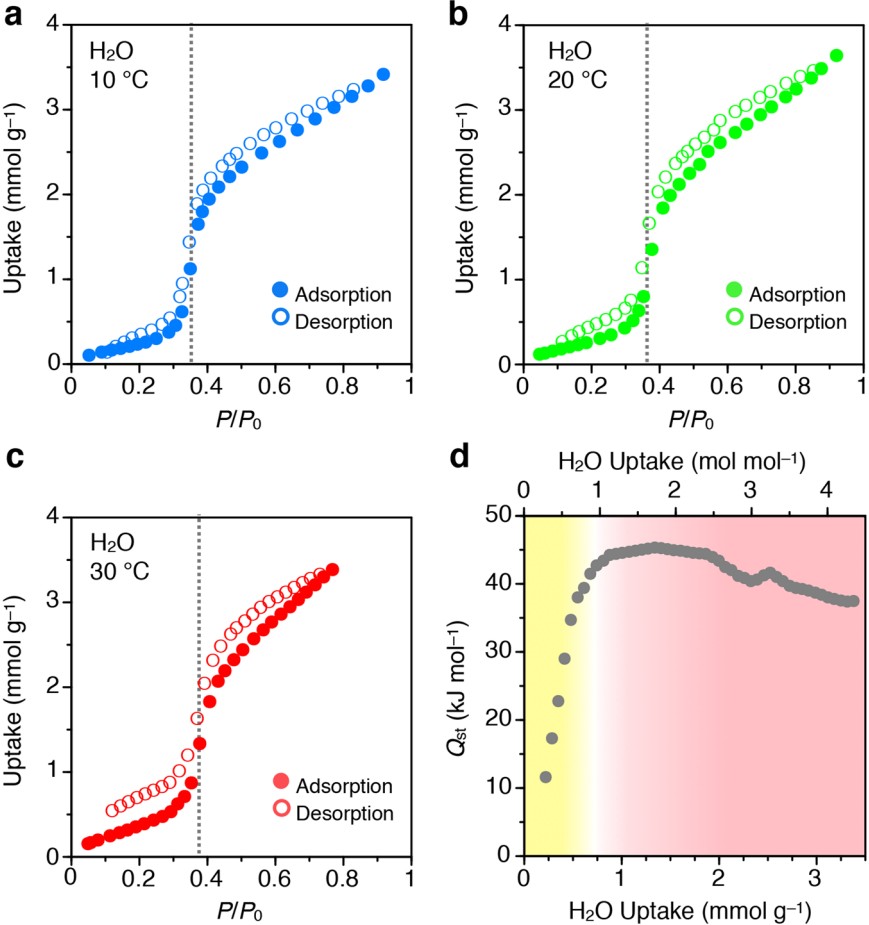

**Fig. 4 $H_2O$ sorption into VPC-1. a–c** $H_2O$ adsorption (closed circles)/desorption (open circles) isotherms of **VPC-1** measured at 10 (**a**), 20 (**b**), and 30 °C (**c**). **d** Isosteric heat of adsorption ($Q_{st}$) for $H_2O$ plotted against the $H_2O$ uptake into **VPC-1**. The upper horizontal axis shows the molar ratio of the $H_2O$ uptake to **1**. Yellow- and red-coloured regions indicate hydrophobic and hydrophilic pore surfaces of **VPC-1**, respectively.

molecules can diffuse into the pores. At a certain $H_2O$ pressure (~50%), the external Cz units start to twist so that the central part of Cz can facilitate enhanced dipole–dipole interactions with the guest $H_2O$ molecules. This affinity transition of the pore surface is followed by the increasingly accelerated adsorption of $H_2O$.

In the present work, we report a hydrochromic VDW porous crystal **VPC-1**. The $H_2O$ sorption induces the change in the electronic absorption of the crystalline powder that is readily detectable with the naked eye. Coincidentally, the carbazole units twist simultaneously and reversibly upon uptake and release of guest $H_2O$ molecules. An affinity transition model is proposed for the simultaneous twisting of the external Cz units and the stepped $H_2O$ sorption of **VPC-1** on the basis of the isosteric heat of adsorption. The present study successfully demonstrates that VDW porous crystals can be attractive candidates towards high-performance sensors or adsorbates under water-containing conditions. Although the on-purpose synthesis of VDW porous crystals is still a fundamental challenge, we believe that our findings will lead to the further elaboration of VDW porous crystals, in an analogous way to MOF, COF and HOF chemistry.

## Methods

**Materials**. Commercial reagents were purchased from Sigma-Aldrich, TCI, or Wako Pure Chemical Industries, Ltd. and used as received. MeOH was dried over activated molecular sieves 3 Å. Toluene was purchased as dehydrated grade and purified by passing through a solvent purification system (KOREA KIYON Co., Ltd.). 3,11-Dibromo-dibenzo[a,j]phenazine **2Br-DBPHZ** and the carbazole dendron **CzG2** were prepared according to the previously reported procedures[27–29].

**General procedures**. All reactions were carried out under an atmosphere of nitrogen unless otherwise noted. Melting points were determined on a Stanford Research Systems MPA100 OptiMelt Automated Melting Point System. [1]H and [13]C NMR spectra were recorded on a on a JEOL JMTC-400/54/SS spectrometer ([1]H NMR, 400 MHz; [13]C NMR, 100 MHz) using tetramethylsilane as an internal standard. Infrared spectra for the characterization of **1** were acquired on a SHI-MADZU IRAffinity-1 FTIR spectrometer. Mass spectra and high-resolution mass spectra (HRMS) were obtained on a JEOL JMS-700 mass spectrometer in fast atom bombardment (FAB) mode using nitrobenzyl alcohol (NBA) as the matrix. Preparative gel permeation liquid chromatography (GPC) was performed on a JAI (Japan Analytical Industry) LC–908 instrument with JAIGEL 1H-2H columns and chloroform as an eluent. Products were purified by chromatography on silica gel BW-300 and Chromatorex NH (Fuji Silysia Chemical Ltd.). Analytical thin-layer chromatography (TLC) was performed on pre-coated silica gel glass plates (Merck silica gel 60 F254 and Fuji Silysia Chromatorex NH, 0.25 mm thickness). Compounds were visualized with UV lamp. Electronic absorption spectra were recorded on a JASCO model V-570 UV/VIS/NIR spectrometer. Diffuse reflectance spectra under atmosphere were recorded on a JASCO model V-570 UV/VIS/NIR spectrometer equipped with a JASCO model ISN-470 integrating sphere option. The photoluminescence spectra were recorded on a JASCO model FP-8500 spectrofluorometer. The sorption isotherm measurements for $N_2$ (99.99995%) and $H_2O$ (purified through a Merck Millipore model Direct-Q 3 water purification system) were performed using a Bel Japan, Inc. model BELSORP-max automatic volumetric adsorption apparatus. A known amount (~20 mg) of **VPC-1**, placed in a glass tube, was dried under a reduced pressure at 80 °C for 10 h to remove the included guest molecules. Powder XRD patterns were recorded on a RIGAKU model Miniflex600 diffractometer with a Cu $K_\alpha$ radiation source (40 kV and 15 mA), equipped with a model D/Tex Ultra2-MF high-speed 1D detector. Thermogravimetric analysis (TGA) was conducted on a Seiko Instruments Inc. model EXSTAR7000: TG/DTA7300 at a heating rate of 10 °C min[−1] under constant Ar flow. Differential scanning calorimetry (DSC) traces were measured on a Seiko Instruments Inc. model EXSTAR7000: X-DSC7000 differential scanning calorimeter using unsealed Al sample pans at a heating/cooling rate of 5 °C min[−1] under constant Ar flow.

**Synthesis of 1**. 3,11-bis(9'$H$-[9,3':6',9''-Tercarbazol]-9'-yl)dibenzo[$a,j$]phenazine **1** was prepared through a Pd-catalyzed amination reaction of 3,11-Dibromo-dibenzo [$a,j$]phenazine **2Br-DBPHZ** with dendrons (**CzG2**) as follows. Toluene was degassed through freeze-pump-thaw cycling for 3 times before used. In a glovebox, to a two-necked reaction tube (10 mL) equipped with a three-way stopcock and a magnetic stir bar, was added Pd[P($t$-Bu)$_3$]$_2$ (2.6 mg, 5 mol%), and the tube was closed with a rubber septum. Outside the glovebox, dibromophenazine **2Br-DBPHZ** (43.8 mg, 0.10 mmol), **CzG2** (109.4 mg, 0.22 mmol), K$_2$CO$_3$ (82.9 mg, 0.60 mmol), and toluene (1 mL) were added under a stream of N$_2$ gas at room temperature, and the resulting mixture was stirred under reflux for 24 h. Water (5 mL) was added to the reaction mixture, and the organic layer was extracted with CHCl$_3$ (20 mL × 3). The combined organic extracts were dried over MgSO$_4$, and the solvent was evaporated in vacuo to give the crude product, which was purified by flash column chromatography on NH silica gel (eluent: $n$-hexane/CHCl$_3$ 95:5–8:2), GPC (eluent: CHCl$_3$), and reprecipitation from a two-phase solvent of $n$-hexane/CHCl$_3$ to give **1** as yellow solid (60.2 mg, 47%). Details of the characterization data of **1** are described in the Supplementary Methods and their NMR spectra and high-resolution MS charts are shown in the Supplementary Figs. 1–4.

**Synthesis of VPC-1**. As-synthesised amorphous yellow solid of **1** (10 mg) was suspended in a mixture of CHCl$_3$ and MeOH (3/7 v/v, 10 mL) at 60 °C for 2 h. The mixture was then cooled to 25 °C and stood for 3 days. The colour of the precipitates eventually turned from yellow to red. The precipitates were collected by filtration and dried under a reduced pressure for 24 h to give **VPC-1**.

**Single-crystal X-ray structure analysis**. Yellow crystals of **1** (CHCl$_3$)$_{1.75}$ that were suitable for X-ray analysis were grown by the slow diffusion of MeOH suspension of **VPC-1** into CHCl$_3$ solution of **1** (1 mg/mL) and subsequent incubation at 25 °C. Single-crystal XRD was measured on a Rigaku XtaLAB HiPix-6000 HPC area detector. Considerably large $R$ values are caused by a poor quality of the crystal and reduced number of parameters used for the refinement. Chloroform were disordered. The electron density attributed to some solvent molecules was not modelled due to the severe disorders. Detailed crystallographic information about **1** (CHCl$_3$)$_{1.75}$ is tabulated in Supplementary Table 1. Crystal packing diagrams of **1** (CHCl$_3$)$_{1.75}$ are shown in Supplementary Fig. 5. CIF is available in Supplementary Data 1.

**Humidity dependent-diffuse reflectance spectroscopy**. Humidity dependent-diffuse reflectance measurements were performed with a beamline BL43IR, SPring-8 synchrotron facility (Hyogo, Japan). Humidity dependent-diffuse reflectance spectra were measured by a miniature spectrometer (Ocean Optics, FLAME-S). The light source was a Deuterium Tungsten Halogen Lamp (Ocean Optics, DH-2000). About 3 mg of powder **VPC-1** sample was put on a BaF$_2$ substrate and inserted into the humidity control cell. The cell has a BaF$_2$ window to pass the light (from visible to mid infrared range). The atmosphere in the cell was controlled by a mixing device (RIGAKU model HUM-1E), which mixes N$_2$ gas (99.99%, generated by a KOFLOC model MNT-0.8SI nitrogen gas generator) and water vapour at the specified ratio[36]. The water used in the mixing device was purified through a Millipore model Elix advantage-3 water purifier. The humidity in the cell was monitored by the humidity sensor (Sensirion model HYT271) that is put near the sample. It took about 5 min to control and stabilize the humidity in the cell. The spectra were measured at an equilibrium condition.

**Humidity dependent-FT IR spectroscopy**. Transmittance measurements were performed with a synchrotron FTIR microspectroscopy at a microoptical spectroscopy station in an infrared beamline BL43IR, SPring-8 synchrotron facility (Hyogo, Japan). At the station, an FTIR microspectrometer (BRUKER model HYPERION infrared microscope with model VERTEX70 FTIR spectrometer) was used with the infrared synchrotron radiation. The transmittance spectra in the range from 800 to 4000 cm$^{-1}$ were collected at 25 °C with a resolution of 2 cm$^{-1}$. About 3 mg of powder **VPC-1** sample was put on a BaF$_2$ substrate and inserted into the humidity control cell. The cell has a BaF$_2$ window to pass infrared light. The atmosphere in the cell was controlled by a mixing device (RIGAKU model HUM-1E), which mixes N$_2$ gas (99.99%, generated by a KOFLOC model MNT-0.8SI nitrogen gas generator) and water vapour at the specified ratio[36]. The water used in the mixing device was purified through a Millipore model Elix advantage-3 water purifier. The humidity in the cell was monitored by the humidity sensor (Sensirion model HYT271) that is put near the sample.

**Variable-humidity powder X-ray diffractometry**. Variable-humidity powder X-ray diffractometry (VH-PXRD) was conducted on a RIGAKU Ultima-IV diffractometer with Cu $K_\alpha$ radiation source. The humidity of the sample space was controlled using RIGAKU HUM-1 humidity controller at 22.6 °C.

**Variable-temperature powder X-ray diffractometry**. Variable-temperature powder X-ray diffractometry (VT-PXRD) was conducted on a RIGAKU model Miniflex600 diffractometer with a Cu $K_\alpha$ radiation source (40 kV and 15 mA),

equipped with a model D/Tex Ultra2-MF high-speed 1D detector and a hotplate. The relative humidity was measured at 25 °C.

**Humidity dependent-Raman spectroscopy**. Raman spectra were measured by a laser Raman spectrometer (JASCO model RMP-335). The wavelength of the laser was 785 nm. About 3 mg of the powder **VPC-1** sample was put on a BaF$_2$ substrate and inserted into the same humidity control cell as was used for infrared spectroscopy. The Raman spectra were collected with a resolution of 2.4 cm$^{-1}$ at 25 °C.

**DFT calculations**. All calculations were performed within the framework of the density functional theory (DFT) using the Gaussian 16 program[32]. DFT calculations were carried out in a gas phase to obtain the optimized structures and the electronic properties of the ground state of **1** using CAM-B3LYP[30], M06-2X[37] and ωB97XD[38] functionals and the 6-31G(d,p)[31] basis set. CAM-B3LYP functional was found to provide a better quantitative agreement with the experimental data compared to other functionals. These results confirm that CAM-B3LYP functional seems appropriate for evaluating the structural and vibrational properties of **1** system. In good agreement with the experimental crystal structure, the DFT-optimized molecular structure of **1** predicts a dihedral angle ($\theta_1$) between the internal Cz units and the central DBPHZ group of 55°, while the dihedral angles between the internal and external Cz ($\theta_2$) is predicted to be 60° (Supplementary Fig. 22). In addition to the optimized molecular structure, three different theoretical models for the **1** system have been studied in order to explain the synchronous flipping from 90 to 60° of the external Cz units regarding to the internal Cz units in response to humid atmosphere (Supplementary Fig. 19). The vertical electronic excitation energies were computed using the time-dependent DFT (TD-DFT) method[39,40]. Absorption spectra were simulated through convolution of the vertical transition energies and oscillator strengths with Gaussian functions characterized by a half width at half-maximum (fwhm) of 0.3 eV. Molecular orbital contours were plotted using GaussView 5.0.

## Data availability
All data that support the findings in this study are available within the article and its Supplementary Information and/or from the corresponding authors on reasonable request. The X-ray crystallographic coordinates for structures reported in this Article have been deposited at the Cambridge Crystallographic Data Centre (CCDC), under deposition number CCDC 2015162. These data can be obtained free of charge from The Cambridge Crystallographic Data Centre via http://www.ccdc.cam.ac.uk/data_request/cif.

## Code availability
The Gaussian 16 package, Revision A.03 was employed for all the DFT calculations[32]. The software and instructions for its use are available at http://gaussian.com/. For the visualization of the vibrational normal modes, the chemcraft program was used, which is available at https://www.chemcraftprog.com.

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

## Acknowledgements

This work was supported by a Grant-in-Aid for Scientific Research on Innovative Areas "π-System Figuration" (JP17H05142, JP17H05155, JP17H05146), "Aquatic Functional Materials" (JP19H05716, JP19H05717), "Coordination Asymmetry" (JP16H06521), Scientific Research (A) (JP16H02081, JP20H00369), Scientific Research (B) (JP19H02746, JP20H02801), Scientific Research (S) (JP15H05757), Joint International Research (JP15KK0182), and Young Scientists (JP19K15334) from Japan Society for the Promotion of Science (JSPS), Leading Initiative for Excellent Young Researchers, MEXT, Kato Foundation for Promotion of Science, The Kao Foundation for Arts and Sciences, Asahi Glass Foundation, Ogasawara Foundation, University of Tsukuba Pre-strategic initiative "Ensemble of light with matters and life", TIA Kakehashi, Cooperative Research Program NJRC Mater. & Dev. Synchrotron radiation measurements were performed at SPring-8 with the approval of JASRI (2018B1278). The work at the University of Malaga was funded by MICNN (PID2019-110305GB-100) and Junta de Andalucia (UMA18-FED-ERJA-080, P09FQM-4708) projects. The authors acknowledge the computer resources, technical expertise, and assistance provided by the SCBI (Supercomputing and Bioinformatics) centre of the University of Malaga.

## Author contributions

H.Y., Y.T., K.A. and Y.Y. designed the experiments. H.Y., S.N., J.Y. and Y.T. performed crystal synthesis and characterizations. M.O., Y.T., S.M., K.A. and K.Y. conducted organic synthesis. H.S. conducted single-crystal X-ray structure analysis. Y.T. and Y.I. conducted humidity-dependent diffuse-reflectance, IR, and Raman spectroscopic measurements. M.C.R.D., M.M.O. and I.B.D. carried out DFT calculations. K.I., K.N., H.T., and S.O. performed VH-PXRD measurements. H.Y. and Y.Y. prepared the manuscript with feedback from the other authors.

## Competing interests

The authors declare no competing interests.
