## [Peer Review File · Communications Chemistry]

Reviewers' comments:

Reviewer #1 (Remarks to the Author):

They synthesized HFC-1 consists of a planar dibenzo[a,j]phenazine (DBPHZ) core and two branched carbazole (Cz) dendrons. The HFC-1 is porous and they discuss the Cz parts flip upon guest insertion. Construction of the stable and dynamic porous molecular architecture with use of single dendrimer molecules 1 is noteworthy. The work would contribute to the further designs of this sort of assembly by functionalization of external moieties of the dendrimers.

One issue is that the crystal structure of HFC-1 is not identified. Schemes in Figure 1a is fully in imagination, not recommended. The "flipping" motion of Cz units were characterized by IR, Raman as well as DFT studies. It is no doubt the Cz units show rearrangement upon guest accommodation. Meanwhile, the term "flipping" is confusing for us. "Flipping" reminds us the dynamic event with some frequency as butterfly wings. All the studies in here is static information-based, and the observation is a solid-to-solid transformation. To monitor the (potential) dynamic behavior of the Cz rings, I suggest DSC or PXRD under VT conditions (without guests), or solid-state NMR, to interpret how the Cz represent an intrinsic flipping. This sort of porous, molecular assembly often show thermal expansion/contraction accompanied with dynamics change of local structure.

Since they do not know the crystal structure, it is hard to distinguish whether (1) the rearrangement of Cz moieties occur by the direct interaction of Cz and H₂O, or, (2) H₂O access to the solid interface to invoke structure rearrangement first, and the rearrangement of Cz occur as a second step. There is no information about the direct contact of guests and Cz units. Other guest sorption isotherms would help to understand the mechanism.

Other comments are listed in below.

- Better to measure TGA for the sample containing solvents or H₂O to show the structure integrity upon desolvation.

- I suggest to measure variable-humidity-PXRD to observe the process of structural change as a function of RH.

- Page 15 "At a certain H₂O pressure (~50%), the external Cz units start to flip so that the central pyrrole moiety can facilitate enhanced dipole-dipole interactions with the guest H₂O molecules. This affinity transition of the pore surface is followed by the increasingly accelerated adsorption of H₂O."

We could assume some change of hydrophilicity as H₂O accommodation, but it is not evident in here. We do not observe a hysteresis in desorption profiles suggesting the interaction of H₂O and "hydrophilic" pore interior is so weak. Can we regard the pores are hydrophilic?

- Single crystal structure of 1 was collected at 93K, and the R1 value is too high (0.16). Is this because of disorder?

The work successfully demonstrated to show the potential of dendrimer-based new host compounds. They carried out reliable and careful characterization for structure, guest responsiveness by various spectroscopies and other physical measurements. With considering they would do major revision, I recommend for publication.

Reviewer #2 (Remarks to the Author):

The paper under review describes the design of hemi-flexible microporous molecular crystals HFC-1, which were formed by thermal annealing of the amorphous dendrimer 1 in the solvent mixture and exhibited colour change in response to a moisture exposure. The authors postulate simultaneous flipping of the outmost carbazole units of dendrimer 1, which alters the hydrophilicity of the pore surface, leading to a significant colour change and sudden H₂O uptake/release without noticeable hysteresis. The flipping of the carbazole unit has been confirmed with the humidity

dependant FTIR spectroscopy and DFT calculation.

Design of the hemi-flexible van der Waals porous crystal framework is really important area of research, which can have a significant impact on materials science in the field of gas storage/release, separation and sensing. The results of the paper can be published after resolving some issues.

- The authors managed to resolve the crystal structure of 1 prepared “the slow diffusion of MeOH suspension of 1 into CHCl₃ solution of 1 (1 mg mL⁻¹) ...” Did they really prepared suspension of 1 in MeOH or the authors meant that “Single crystals of 1 were grown by the slow diffusion of MeOH into CHCl₃ solution of 1 (1 mg mL⁻¹) and subsequent incubation at 25 °C”, which makes more sense. The crystal growth should be described more carefully for the sake of reproducibility if such is required.
- On the other hand, the authors were unable to resolve the structure of HFC-1 “due to its high polycrystallinity”. In this respect, the thermal morphological transitions of HFC-1 are really important not only for its full characterisation but also for estimating the scope of possible applications of this microporous material. Both TGA and DSC are required for HFC-1 and the crystalline material with the solved structure as a comparison.
- Certainly, if the authors demonstrated humidity sensor on the basis of HFC-1 it would be ideal for publication in such high impact journal as Science. However, given the information provided in the manuscript it can be published as it is, provided that the authors clarify the most important experiment describing sigmoidal H₂O sorption, the diffuse reflectance spectroscopy. The author should more describe the control of humidity inside integrating sphere during the reflectance spectra measurements. Was it the same setup as was used for FTIR measurements? How quick the sample reach the equilibrium upon changing the humidity?
- The reflectance spectra of HFC-1 immersed in the different solvent (Figure S4 (a)) should be shown in a wider wavelength range to reveal clearly onset for all spectra.
- The error on the figure S15 should be corrected. The charge on the phenylene units of carbazole are positive (not -0.238e, but +0.238e) to provide the electro-neutrality for the whole molecule.

Reviewer #3 (Remarks to the Author):

The manuscript proposed here is well written and well-argumented.

The conclusions are supported by conving results obtained from complementary experiments and calculations.

I strongly recommend the publication of this work

Point-by-Point **Answers** to Reviewers' Comments

Answers to the Comments by Reviewer 1

1. They synthesized HFC-1 consists of a planar dibenzo[a,j]phenazine (DBPHZ) core and two branched carbazole (Cz) dendrons. The HFC-1 is porous and they discuss the Cz parts flip upon guest insertion. Construction of the stable and dynamic porous molecular architecture with use of single dendrimer molecules 1 is noteworthy. The work would contribute to the further designs of this sort of assembly by functionalization of external moieties of the dendrimers.

=> We appreciate the reviewer for the highly encouraging comments.

2. One issue is that the crystal structure of HFC-1 is not identified. Schemes in Figure 1a is fully in imagination, not recommended.

=> The comment by the reviewer (and the editor) on the lack of the crystal structure is reasonable. However, in spite of our enormous efforts to synthesize a single crystal of **VPC-1**, it is quite difficult to obtain high quality ones enough for single crystal XRD analysis. Therefore, we decide to revise the introductory part of the paper. In the revised manuscript, we emphasize the steep and sigmoidal hydrochromism of organic van der Waals (VDW) crystal by the synchronous twisting of the Cz units without change of the crystal lattice. In particular, we highlight that this is the first report on hydrochromism of VDW molecular porous crystals. Accordingly, Figure 1 is properly revised without drawing the molecular rotation in the lattice. With these revisions, the significance of our research is now based on the chromic and water sorption performances of VDW porous crystal. We now believe that the lack of single-crystal data is not as critical as it was in the original version.

Figure 1. | Schematic representations of the hydrochromism of VPC-1 upon hydration/dehydration. **a**, Photographs of VPC-1^{red} and VPC-1^{yellow} together with schematic illustrations of the microporous crystalline grains that vary colours in response to H₂O vapour. **b**, Molecular structure of a second-generation dendrimer 1 with two branched Cz dendrons and a DBPHZ core.

=> We modified the title of the paper and the nomenclature of the compounds to make the whole text coherent. The revised title is “*Sigmoidally hydrochromic molecular porous crystal with rotatable dendrons*”. We also renamed **HFC-1** in the original manuscript to **VPC-1** in the revised manuscript (VPC represents “van der Waals porous crystal”).

3. The "flipping" motion of Cz units were characterized by IR, Raman as well as DFT studies. It is no doubt the Cz units show rearrangement upon guest accommodation. Meanwhile, the term "flipping" is confusing for us. "Flipping" reminds us the dynamic event with some frequency as butterfly wings.

=> According to the reviewer's comment, we replaced the word “flipping” with “twisting” for better readability.

4. All the studies in here is static information-based, and the observation is a solid-to-solid transformation. To monitor the (potential) dynamic behavior of the Cz rings, I suggest DSC or PXRD under VT conditions (without guests), or solid-state NMR, to interpret how the Cz represent an intrinsic flipping. This sort of porous, molecular assembly often show thermal expansion/contraction accompanied with dynamics change of local structure.

=> We highly appreciate the reviewer for the important comments. We newly conducted variable temperature (VT)- and variable humidity (VH)-PXRD measurements using the powder sample of **VPC-1**. The VT-PXRD pattern exhibited an abrupt and reversible enhancement/attenuation of some diffraction peaks upon heating and cooling cycles in a temperature range from 25 to 45 °C. The threshold temperature increased in accordance with the increase of the relative humidity of the surrounding atmosphere (Supplementary Figs. 10–12). Meanwhile, no noticeable peak shift was observed upon the heating/cooling cycle. An analogous abrupt peak enhancement/attenuation without peak shift was observed in VH-PXRD measurements at 22.6 °C (Supplementary Fig. 9), whose threshold is consistent with those in the VT-PXRD measurements. The absence of peak shift supports our claim that the crystalline framework is static. The change in peak area is attributed to the thermal disorder of the Cz moiety, which is also in line with our original claim. Typically, the diffraction peak area represents the crystallinity of the grain or the thermal disorder of the constituent molecules. Therefore, the attenuation of peak area upon cooling is likely attributed to the enhanced thermal disorder of the Cz moiety caused by the adsorption of the H₂O molecules.

=> Another important finding from these experiments is that the VH- and VT-PXRD profiles allow us to draw a phase diagram between the **VPC-1^{yellow}** and **VPC-1^{red}** as a function of relative humidity and temperature (Supplementary Fig. 13).

⇒ We newly added to the revised main text the following sentences: “We conducted variable-humidity (VH) and variable-temperature (VT) PXRD measurement with VPC-1 to evaluate the detailed structural change upon H₂O sorption. The VH-PXRD patterns at 22.6 °C exhibited an abrupt and reversible enhancement/attenuation of the peak area upon increasing and decreasing the humidity with a threshold at around 40 %RH, while no significant peak shift was observed (Supplementary Fig. 9). The unchanged peak positions indicate that the crystalline lattice of VPC-1 is static upon H₂O sorption, while the enhancement/attenuation of the peak area indicates the decreased/increased degree of the thermal disorder of 1. Analogously, VT-PXRD patterns, measured in an unsealed chamber under atmosphere with RH of 69 % at 25 °C, exhibited abrupt and reversible peak enhancement and attenuation with a threshold at around 33 °C without any noticeable peak shift upon heating and cooling (Supplementary Fig. 10). The threshold temperatures decreased as the humidity of the surrounding atmosphere decreased (Supplementary Figs. 11 and 12). The elevated temperature triggers the decrease of the RH in the sample chamber and the release of H₂O molecules from VPC-1^{red}, leading to the enhancement of the peak intensity in the PXRD profiles. Based on the VH- and VT-PXRD results, a relative humidity–temperature phase diagram for VPC-1^{yellow} and VPC-1^{red} is depicted (Supplementary Fig. 13).” (main text, page 7, line 13–page 8, line 5)

Supplementary Fig. 9. (a, b) VH-PXRD profiles of VPC-1 upon (a) drying and (b) moistening measured under sealed chamber at constant temperature of 22.6 °C. (c, d) Magnified VH-PXRD profiles of Supplementary Fig. 9a and b. (e) A plot of the peak areas of the diffraction peak at $2\theta \sim 8^\circ$ against the relative humidity.

Supplementary Fig. 10. (a, b) VT-PXRD profiles of VPC-1 upon (a) heating and (b) cooling measured under atmosphere with relative humidity of 69 %RH at 25°C. (c) A plot of the peak areas of the diffraction peak at 2θ of 8.1° against the temperature. (d) Photographs of the powder samples of VPC-1 at 25 (upper) and 45 °C (bottom).

Supplementary Fig. 11. (a, b) VT-PXRD profiles of VPC-1 upon (a) heating and (b) cooling measured under atmosphere with relative humidity of 58 %RH at 25°C. (c) A plot of the peak areas of the diffraction peak at 2θ of 8.1° against the temperature. (d) Photographs of the powder samples of VPC-1 at 25 (upper) and 45 °C (bottom).

Supplementary Fig. 12. (a, b) Variable-temperature powder X-ray diffraction (VT-PXRD) profiles of VPC-1 upon (a) heating and (b) cooling measured under atmosphere with relative humidity of 47 %RH at 25°C. (c) A plot of the peak areas of the diffraction peak at 2θ of 8.1° against the temperature. (d) Photographs of the powder samples of VPC-1 at 25 (upper) and 45 °C (bottom).

Supplementary Fig. 13. A phase diagram of VPC-1 plotted based on the results of VH-PXRD (Supplementary Fig. 9) and VT-PXRD (Supplementary Figs. 10–12). The red and yellow markers indicates VPC-1^{red} and VPC-1^{yellow}, respectively. The pentagon markers represent the VH-PXRD experiments. The circle, triangle and square markers represent the VT-PXRD experiments measured under atmosphere with relative humidity at 25°C of 47, 58 and 69 %, respectively. The relative humidity at given temperature is mathematically calculated based on the reported saturation pressure of water. The plausible phase area of VPC-1^{red} and VPC-1^{yellow} is depicted as red and yellow background, respectively.

=> We newly conducted DSC measurements with **VPC-1**. The DSC charts are featureless in the measured temperature range (30–200 °C). This is because all the H₂O molecules adsorbed into **VPC-1** were totally removed by the Ar flow during the temperature stabilization process. We newly added to the revised SI the DSC charts as Supplementary Fig. 8.

Supplementary Fig. 8. Differential scanning calorimetric profiles of **VPC-1** upon heating (orange curve) and cooling (blue curve) at a rate of 5 °C min⁻¹ under constant Ar flow.

5. Since they do not know the crystal structure, it is hard to distinguish whether (1) the rearrangement of Cz moieties occur by the direct interaction of Cz and H₂O, or, (2) H₂O access to the solid interface to invoke structure rearrangement first, and the rearrangement of Cz occur as a second step. There is no information about the direct contact of guests and Cz units. Other guest sorption isotherms would help to understand the mechanism.

=> Transmission of the molecular conformational motion always involves the distortion of the crystal lattice. However, in the present case, the crystalline lattice is kept intact during the H₂O sorption as revealed by the static PXRD peaks upon H₂O sorption. Moreover, as described in the main text in regard to the FTIR and Raman spectra (Figs. 3 and S16), the external Cz units are solely mobile during the H₂O sorption. These results corroborate the hypothesis (1) and rebut the hypothesis (2). We newly added to the revised manuscript a description as follows: “Taking this fact and the results of PXRD measurements, we conclude that the external Cz units are solely responsive and mobile in **VPC-1^{yellow}** and **VPC-1^{red}**, while the other moiety of **1** and the crystal packing mode are static in the H₂O sorption process. This is in clear contrast with the conventional flexible porous crystals, which transform the whole crystalline lattices and the conformation of the constituent molecules simultaneously upon guest sorption.”. (main text, page 10, lines 4–8)

=> As the reviewer 1 pointed out, there is no direct or crystallographic evidence for the physical contact of guest H₂O molecules with the external Cz, although the hypothesis (1) is indirectly but consistently supported by our spectroscopic results. To circumvent this issue, we revised the whole story of the manuscript as written in the above (item 2). In the revised version, the significance and the novelty are put on the finding of the first hydrochomic behavior with VDW porous crystal in addition to the twisting of the Cz units upon H₂O sorption.

6. Better to measure TGA for the sample containing solvents or H₂O to show the structure integrity upon desolvation.

=> As described in the original manuscript (page 9, lines 9–13) and in the revised manuscript (page 6, lines 15–16), the powder specimen of VPC-1 was pre-dried under reduced pressure before experiments, corroborating the structural integrity of VPC-1 upon desolvation. We also conducted TG analysis for red-colored powder of VPC-1 containing H₂O molecules. However, the profile was almost featureless in the measured temperature range (40–200 °C). This is most likely because all the adsorbed H₂O molecules were removed just by the Ar flow during the temperature stabilization process.

Supplementary Fig. 7. Thermogravimetric analysis of VPC-1 upon heating at a heating rate of 5 °C min⁻¹ under constant Ar flow.

=> For better readability about the desolvation of VPC-1, we added to the revised manuscript the following sentence: “The complete desolvation of VPC-1 was further confirmed by thermogravimetric analysis and differential scanning calorimetry measurement under constant Ar flow condition (Supplementary Figs. 7 and 8).” (main text, page 6, lines 19–21)

7. I suggest to measure variable-humidity-PXRD to observe the process of structural change as a function of RH.

=> We appreciate the reviewer for the valuable comment. We newly conducted the VH-PXRD. The detailed experimental results and the conclusions are written in the above (item 4).

8. Page 15 "At a certain H₂O pressure (~50%), the external Cz units start to flip so that the central pyrrole moiety can facilitate enhanced dipole–dipole interactions with the guest H₂O molecules. This affinity transition of the pore surface is followed by the increasingly accelerated adsorption of H₂O." We could assume some change of hydrophilicity as H₂O accommodation, but it is not evident in here. We do not observe a hysteresis in desorption profiles suggesting the interaction of H₂O and "hydrophilic" pore interior is so weak. Can we regard the pores are hydrophilic?

=> The absence of hysteresis is consistent with the “hydrophilicity” of the pore surface. This is because the pore space in VPC-1 is very small less than 1 nm in diameter (see Supplementary Fig. 14), to prevent the H₂O molecules from the capillary condensation

therein. Previous researches summarize the experimental results and theoretical explanations for non-hysteretic H₂O sorption in “hydrophilic” pores with diameters less than 2.0 nm (“Such a non-hysteretic sorption behaviour indicates the reversible and continuous pore filling by H₂O molecules rather than a capillary condensation^{15,33}.” (main text, page 11, lines 6–8). Therefore, we evaluate the hydrophilicity/hydrophobicity of the pore surface based on the heat of adsorption, as described in the main text “the Q_{st} plot exhibits a plateau with Q_{st} of 44 ± 1 kJ mol⁻¹, which is a typical value for H₂O physisorption to the hydrophilic surface.” (main text, page 12, lines 6–8).

9. Single crystal structure of **1** was collected at 93K, and the R1 value is too high (0.16). Is this because of disorder?

=> We appreciate the comment. As the reviewer commented, the severe disorder of the solvent molecules affected the *R* values. The previously submitted cif file contained a description about the quality of the data as follows: “Considerably large *R* values might be due to a poor quality of the crystal and reduced number of parameters used for the refinement. Chloroform were disordered. The electron density attributed to some solvent molecules was not modelled due to the severe disorders”. We added to the revised SI the same description for better understanding of the readers. (SI, page S8, lines 3–5)

=> We added crystal packing diagram of **1** for better understanding of the readers (Supplementary Fig. 3).

Supplementary Fig. 3. Crystal packing diagrams of **1**(CHCl₃)_{1.75}, viewed along the crystallographic *b*, *c*, and *a* axes directions (**a**, **b**, and **c**, respectively). The hydrogen atoms and guest CHCl₃ molecules are omitted for clarity. Symmetrically equivalent molecules are depicted in the same colour.

Answers to the Comments by Reviewer 2

1. The paper under review describes the design of hemi-flexible microporous molecular crystals **HFC-1**, which were formed by thermal annealing of the amorphous dendrimer **1** in the solvent mixture and exhibited colour change in response to a moisture exposure. The authors postulate simultaneous flipping of the outmost carbazole units of dendrimer **1**, which alters the hydrophilicity of the pore surface, leading to a significant colour change and sudden H₂O uptake/release without noticeable hysteresis. The flipping of the carbazole unit has been confirmed with the humidity dependant FTIR spectroscopy and DFT calculation. Design of the hemi-flexible van der Waals porous crystal framework is really important area of research, which can have a significant impact on materials science in the field of gas storage/release, separation and sensing. The results of the paper can be published after resolving some issues.

=> We appreciate the reviewer for the highly encouraging comments.

2. The authors managed to resolve the crystal structure of **1** prepared “the slow diffusion of MeOH suspension of **1** into CHCl₃ solution of **1** (1 mg mL⁻¹) ...” Did they really prepared suspension of **1** in MeOH or the authors meant that “Single crystals of **1** were grown by the slow diffusion of MeOH into CHCl₃ solution of **1** (1 mg mL⁻¹) and subsequent incubation at 25 °C”, which makes more sense. The crystal growth should be described more carefully for the sake of reproducibility if such is required.

=> We do understand the suspicion from the reviewer about the preparation method for single crystal. However, the method is actually what we applied. As described in the revised manuscript, **VPC-1** formed only after the thermal annealing of the amorphous precipitation of **1** in the mixture of MeOH and CHCl₃. We found that analogous transformation also occurred when incubating the MeOH suspension of **VPC-1** at 25 °C upon slow diffusion of CHCl₃ solution of **1**. To make the synthetic procedure clearer for readers, we added to the main text a description as follows: “*Single crystals of **1** were grown as follows: A MeOH layer containing dispersed solid powder of **1** was gently put on a CHCl₃ solution of **1** (1 mg mL⁻¹). The container was incubated for three days at 25 °C to yield yellow-coloured platelet single crystals with side length of several tens of micrometres*” (main text, page 5, lines 10–13).

3. On the other hand, the authors were unable to resolve the structure of **HFC-1** “due to its high polycrystallinity”. In this respect, the thermal morphological transitions of **HFC-1** are really important not only for its full characterisation but also for estimating the scope of possible applications of this microporous material. Both TGA and DSC are required for **HFC-1** and the crystalline material with the solved structure as a comparison.

=> We appreciate the reviewer for the important comments. We newly conducted TGA (40–200 °C) and DSC (30–200 °C) measurements for **VPC-1** (Supplementary Figs. 7 and 8). Both charts are substantially featureless, indicating that neither noticeable weight loss nor structural transition take place in **VPC-1** in this temperature range. Unfortunately, the TG and DSC measurements for crystalline material with the solved structure was unsuccessful because the material is always obtained as a mixture with **VPC-1**.

Supplementary Fig. 7. Thermogravimetric analysis of **VPC-1** upon heating at a heating rate of 5 °C min⁻¹ under constant Ar flow.

Supplementary Fig. 8. Differential scanning calorimetric profiles of **VPC-1** upon heating (orange curve) and cooling (blue curve) at a rate of 5 °C min⁻¹ under constant Ar flow.

- Certainly, if the authors demonstrated humidity sensor on the basis of **HFC-1** it would be ideal for publication in such high impact journal as Science. However, given the information provided in the manuscript it can be published as it is, provided that the authors clarify the most important experiment describing sigmoidal H₂O sorption, the diffuse reflectance spectroscopy. The author should more describe the control of humidity inside integrating sphere during the reflectance spectra measurements. Was it the same setup as was used for FTIR measurements? How quick the sample reach the equilibrium upon changing the humidity?

=> We appreciate the reviewer for the comment. The setup is almost identical with that for the VH-FTIR measurement. We newly added to the revised SI about the detailed description about the humidity control setups for the diffusion reflectance and FTIR measurements as follows: “Humidity dependent-diffuse reflectance spectra were measured by a miniature spectrometer (Ocean Optics, FLAME-S). The light source was a Deuterium Tungsten Halogen Lamp (Ocean Optics, DH-2000). About 3 mg of powder **VPC-1** sample was put on a BaF₂ substrate and inserted into the humidity control cell. The cell has a BaF₂ window to pass infrared light. The atmosphere in the cell was controlled by a mixing device

(RIGAKU model HUM-1E), which mixes N_2 gas (99.99%, generated by a KOFLOC model MNT-0.8SI nitrogen gas generator) and water vapour at the specified ratio^{S3}. The water used in the mixing device was purified through a Millipore model Elix advantage-3 water purifier. The humidity in the cell was monitored by the humidity sensor (Sensirion model HYT271) that is put near the sample. It took about 5 minutes to control and stabilize the humidity in the cell. The spectra were measured at an equilibrium condition.” (SI, page S11, lines 1–11)

=> The color change occurs within a few seconds after the humidity or temperature goes across the threshold. However, the spectroscopies we applied (diffuse reflectance, PXRD, IR, Raman) are not quick enough to follow the kinetics. Alternatively, we took a movie on the colour change during the reduction process of pressure (Supplementary Movie 1). We newly added to the main text a description as follows: “The colour change between *VPC-I^{yellow}* and *VPC-I^{red}* was completed within ten seconds upon drying under reduced pressure, which is much less than the interval time required for the stabilization of humidity or temperature (Supplementary Movie 1).” (main text, page 7, lines 9–12)

Supplementary Movie 1. A movie recording the colour change of *VPC-I^{red}* from red to yellow upon drying by reducing pressure inside a glass container. The colour change was completed within 10 sec after starting the vacuuming.

5. The reflectance spectra of **HFC-1** immersed in the different solvent (Figure S4 (a)) should be shown in a wider wavelength range to reveal clearly onset for all spectra.

=> We replaced the diffuse reflectance spectra with newly measured ones, featuring a wider wavelength range (400–900 nm, Supplementary Fig. 5). We also added Tauc plots utilized for the calculation of the band gaps.

Supplementary Fig. 5. (a, b) Diffuse reflectance spectra (a) and Tauc plots (b) of VPC-1 immersed in hexane (blue curve), 2-propanol (green curve), MeOH (orange curve), and H₂O (red curve). (c) The E_{T30} plot of VPC-1 immersed in hexane (blue circle), 2-propanol (green circle), MeOH (orange circle), and H₂O (red circle).

6. The error on figure S15 should be corrected. The charge on the phenylene units of carbazole are positive (not $-0.238e$, but $+0.238e$) to provide the electro-neutrality for the whole molecule.

⇒ We appreciate the reviewer for pointing out the typo. We revised the values in Supplementary Fig. 23 according to the reviewer's comment.

Supplementary Fig. 23. DFT-calculated atomic charge distribution of carbazole at the CAM-B3LYP/6-31G** level.

Answer to the Comment by Reviewer 3

1. The manuscript proposed here is well written and well-argumented. The conclusions are supported by conving results obtained from complementary experiments and calculations. I strongly recommend the publication of this work

=> We highly appreciate the reviewer for this encouraging comment.

REVIEWERS' COMMENTS:

Reviewer #1 (Remarks to the Author):

Through the revision process, they conducted almost all the suggested experiments to figure out the dynamic property of the compounds upon heating/guest sorption. The observed spectroscopies strengthen to visualize the mobility and structural transformation which is a main part of manuscript. As a result, the revised manuscript became more solid in discussion. They also improved crystal structure data-set, and provided better figures as well as changing title/introduction significantly to reply reviewers' comments.

With their large efforts on modification of manuscript with a number of new experiments, I think the current version has sufficient novelty and convincing discussion, therefore I recommend for publication.

Reviewer #2 (Remarks to the Author):

The paper after revision looks better all comments have been addressed.

The only issue which still need to be resolved is TGA data for the final material. The TGA curve looks a little bit slanted with weight of the sample decreasing steadily. According to authors the slight drop in the weight of the sample is not due to the loss of water and must be due to a poor calibration. TGA should be performed in a wider range of the temperature up to 500 C at least to register decomposition temperature of the product (5% weight loss T_d which would require more steady horizontal line at the very beginning of the TGA curve, or onset of decomposition, or deflection point). These parameters extracted from TGA are conventional for a standard characterisation of materials and can be useful for estimating the thermal stability of the compound.

Point-by-Point **Answers** to Reviewers' Comments

Answer to the Comment by Reviewer 1

1. Through the revision process, they conducted almost all the suggested experiments to figure out the dynamic property of the compounds upon heating/guest sorption. The observed spectroscopies strengthen to visualize the mobility and structural transformation which is a main part of manuscript. As a result, the revised manuscript became more solid in discussion. They also improved crystal structure data-set, and provided better figures as well as changing title/introduction significantly to reply reviewers' comments.

With their large efforts on modification of manuscript with a number of new experiments, I think the current version has sufficient novelty and convincing discussion, therefore I recommend for publication.

=> We appreciate the reviewer for the highly encouraging comments.

Answer to the Comments by Reviewer 2

2. The paper after revision looks better all comments have been addressed.

=> We appreciate the reviewer for the highly encouraging comments.

3. The only issue which still need to be resolved is TGA data for the final material. The TGA curve looks a little bit slanted with weight of the sample decreasing steadily. According to authors the slight drop in the weight of the sample is not due to the loss of water and must be due to a poor calibration. TGA should be performed in a wider range of the temperature up to 500 C at least to register decomposition temperature of the product (5% weight loss Td which would require more steady horizontal line at the very beginning of the TGA curve, or onset of decomposition, or deflection point). These parameters extracted from TGA are conventional for a standard characterization of materials and can be useful for estimating the thermal stability of the compound.

=> According to the comment by reviewer 2, we newly conducted TG analysis of **VPC-1** in a temperature range from 45 to 500 °C after precise calibration (Supplementary Fig. 9). The

TG curve exhibited no obvious weight loss up to 300 °C. This trend corroborates our claim that **VPC-1** is completely dehydrated under ambient conditions. The thermal decomposition temperature for the 5% mass loss was 461 °C. We replaced the previous TG curve with the newly measured one together with a description about the dehydration and thermal decomposition temperature of **VPC-1** in Supplementary Information (SI, page 11).

Supplementary Fig. 9. Thermogravimetric analysis of **VPC-1** upon heating at a rate of 10 °C min⁻¹ under constant Ar flow. The decomposition temperature for the 5 % ($T_{5\%}$) mass loss is 461 °C.